# Cannabinoid Receptor 2 (CB2) Inverse Agonist SMM-189 Induces Expression of Endogenous CB2 and Protein Kinase A That Differentially Modulates the Immune Response and Suppresses Experimental Colitis

**DOI:** 10.3390/pharmaceutics14050936

**Published:** 2022-04-26

**Authors:** Sonia Kiran, Ahmed Rakib, Bob M. Moore, Udai P. Singh

**Affiliations:** Department of Pharmaceutical Sciences, College of Pharmacy, The University of Tennessee Health Science Center, 881 Madison Avenue, Memphis, TN 38163, USA; skiran@uthsc.edu (S.K.); arakib@uthsc.edu (A.R.); bmoore@uthsc.edu (B.M.M.)

**Keywords:** colitis, cannabinoid receptor 2 (CB2), Th17 cells, macrophages, inflammatory bowel disease

## Abstract

The causes of Crohn’s disease (CD) and ulcerative colitis (UC), the two most common forms of inflammatory bowel disease (IBD), are multi-factorial and include dysregulation of immune cells in the intestine. Cannabinoids mediate protection against intestinal inflammation by binding to the G-protein coupled cannabinoid receptors 1 and 2 (CB1 and CB2). Here, we investigate the effects of the CB2 inverse agonist SMM-189 on dextran sodium sulfate (DSS)-induced experimental colitis. We observed that SMM-189 effectively attenuated the overall clinical score, reversed colitis-associated pathogenesis, and increased both body weight and colon length. Treatment with SMM-189 also increased the expression of CB2 and protein kinase A (PKA) in colon lamina propria lymphocytes (LPLs). We noticed alterations in the percentage of Th17, neutrophils, and natural killer T (NKT) cells in the spleen, mesenteric lymph nodes (MLNs), and LPLs of mice with DSS-induced colitis after treatment with SMM-189 relative to DSS alone. Further, myeloid-derived suppressor cells (MDSCs) during colitis progression increased with SMM-189 treatment as compared to DSS alone or with control cohorts. These findings suggest that SMM-189 may ameliorate experimental colitis by inducing the expression of endogenous CB2 and PKA in LPLs, increasing numbers of MDSCs in the spleen, and reducing numbers of Th17 cells and neutrophils in the spleen, MLNs, and LPLs. Taken together, these data support the idea that SMM-189 may be developed as a safe novel therapeutic target for IBD.

## 1. Introduction

Ulcerative colitis (UC) and Crohn’s disease (CD), two forms of inflammatory bowel disease (IBD), affect nearly 1.4 million people in the USA and represent a rising challenge to human health around the world [1]. The exact etiology of IBD remains a conundrum for the scientific community; however, it is clear that dysregulation of the immune response in the presence of an external insult results in damage to the intestinal mucosa, contributing to the development of IBD [2]. Many anti-inflammatory agents and immunosuppressants are in use in the clinic to stem the severity of IBD flares, but these agents can have serious side effects and some patients’ symptoms are refractory to them [3]. For this reason, many IBD patients can be resistant to treatment, which justifies the continued search for natural, safe, and effective new therapeutic approaches. To this end, nearly 50% of IBD patients are treated with alternative medications like probiotics, prebiotics, and medical cannabis [4].

A diverse population of immune cells resides in the inflamed gut of an IBD patient, but T cells, macrophages, and natural killer T (NKT) cells act as the main pathogenic immune cells mediating inflammation in intestinal tissue [5,6]. In various models of IBD, T helper 17 (Th17) cells that produce interleukin-17 (IL-17) represent one of the most abundant cell types in the inflamed colon, underlying their contribution to IBD development [7]. Myeloid-derived suppressor cells (MDSCs) represent a mixed population of immature myeloid cells that serve as potent suppressors of the immune response [8]. In chronic inflammatory pathologies, MDSCs expand and serve as crucial players in the prevention and treatment of human diseases [9]. The numbers of MDSCs dramatically increase during intestinal inflammation in mice, where they suppress IFN-γ release by T cells [10]. During the progression of colitis, neutrophils secrete both pro-inflammatory and anti-inflammatory cytokines [11], including IL-17 [12], whose association with colitis suggests that IL-17 produced by neutrophils plays a major role in gut inflammation. Neutrophil-derived IL-17 has also been implicated in murine kidney ischemia-reperfusion injury, suggesting that its production by neutrophils in the intestine may have a role in colon inflammation [13]. Detrimental immune responses are also mediated by NKT cells that play both protective and destructive roles in the various models of experimental colitis. Taken together, these studies suggest that in DSS-induced colitis, the functions of, and crosstalk among, various immune cell populations including Th17 cells, MDCSs, neutrophils, and NKT cells play key roles in maintaining intestinal immune inflammation during colitis. 

For a long time, people have used cannabidiol and other non-psychoactive cannabinoids for the prevention and treatment of gastrointestinal disorders [14]. Emerging evidence reveals that cannabinoid receptor type-2 (CB2) serves as an ideal therapeutic agent that can suppress the severity of IBD [15,16,17]. Further, it has been shown that an active cannabis extract with high cannabidiol attenuates chemically induced intestinal inflammation and hypermotility in the mouse [18]. Interestingly, CB2 receptor agonists can regulate the anti-inflammatory responses of intestinal epithelial cells in experimental animal models of colitis [19]. In vivo studies of CB1 and CB2 receptor activation reveal suppression of proinflammatory cytokine expression, which emphasizes its importance in protection against experimental colitis [20]. Towards this, the selective CB2 agonist JWH-133 reduced inflammation in DSS-induced colitis [21], and another agonist (AM-1241) exhibited anti-inflammatory properties in colitis models [22]. However, to the best of our knowledge, very little mechanistic information is available on the effect of CB2 inverse agonists on experimental colitis in mice. Therefore, in this study, we aimed to understand the cellular and molecular mechanism of a potent CB2 inverse agonist SMM-189 in a DSS-induced experimental model of colitis. The results of our experiments show that SMM-189 increased the expression of endogenous CB2 receptors and PKA in LPLs, reduced numbers of Th17 cells and neutrophils in the spleen, and LPLs, and increased numbers of MDSCs and NKT cells. 

## 2. Material and Methods

### 2.1. Animals

Female C57BL/6 mice, aged ~9 weeks and weighing 18–20 g, were purchased from Jackson Laboratories (Bar Harbor, ME). All mice were maintained in isolator cages and normal light and dark cycles (12:12 h) for a week of acclimatization to the conventional housing conditions of the UTHSC animal facility to minimize animal pain and distress. Following a week of acclimatization, the mice were randomly divided into three groups (control, DSS, and SMM-189) consisting of six each (total *n* = 18) for experimental studies that were repeated three times (*n* = 54). This animal experimentation was performed under a protocol (UPS# 20-0169) approved by the University of Tennessee Health Science Center (UTHSC) Institutional Animal Care and Use Committee (IACUC).

### 2.2. DSS-Induced Experimental Colitis

DSS experimental colitis was induced as we described previously [23]. At the age of ~10 weeks, mice assigned to the control group were provided normal drinking water (ad libitum), while mice in the dextran sodium sulfate (DSS) and SMM-189 groups were provided drinking water containing 3.5% DSS (MP Biomedical, LLC, OH; molecular weight 36,000–50,000) for 7 days (ad libitum) followed by normal water for one day. The body weight, stool consistency, and animal behaviors were recorded daily. All efforts were maintained to reduce the possibility of stress and pain in the mice. We also monitored other symptoms of colitis, including diarrhea, stool consistency, and blood in the fecal matter. At the end of the experimental period, the animals were euthanized. Colon samples were collected, washed with phosphate-buffered saline, cut longitudinally, fixed with formalin, and embedded in paraffin.

### 2.3. Pharmacological Treatments of SMM-189

CB2 inverse agonist SMM-189 was synthesized and formulated by our collaborator Dr. Bob M. Moore in the Department of Pharmacy at UTHSC. Vehicle and SMM-189 formulations were prepared using ethanol: Cremophor: saline (5:5:90) mixture. Mice in the DSS and SMM-189 groups were injected intraperitoneally with 100 µL vehicle (DSS control) or SMM-189 (6.25 mg/kg body weight) each day until the experiment ended on day 8.

### 2.4. Single-Cell Suspensions

At the experimental endpoint, day 9 after treatment, all mice were euthanized with an overdose of inhaled isoflurane. The spleen and mesenteric lymph nodes (MLNs) of each mouse were harvested individually and dissociated using a stomacher (Seward Stomacher^®^ 80 Seward Laboratory Systems Inc. Bohemia, NY, USA) to produce a single-cell suspension. The red blood cells were removed from the spleen with the help of lysis buffer (ThermoFisher Scientific Waltham, MA, USA) and cell debris was removed with a 70 µm cell strainer (Sigma, St. Louis, MO, USA). The resulting cell suspensions were washed with RPMI-1640 medium containing 10% fetal bovine serum (FBS) and placed at 4 °C for analysis by flow cytometry the same day. The cells of the intestinal lamina propria (LPLs) were isolated using a commercial lamina propria dissociation kit (MACS Miltenyi Biotec, 130-097-410 Auburn, CA, USA) with the help of MACS C tubes (MACS Miltenyi Biotec, 130-096-334 Auburn, CA, USA) and a gentleMACS™ Dissociator (MACS Miltenyi Biotec, 130-093-235 Auburn, CA, USA) as recommended by the manufacturer. Lymphocytes were maintained in the complete medium as previously described [21].

### 2.5. Flow Cytometry Analysis

Compensation beads, isotype controls, and fluorescent-conjugated antibodies were purchased from BD and Biolegend (San Diego, CA, USA). The cells isolated from the spleens, MLNs, and LPLs from each group of mice by the above-mentioned protocol were pooled, pelleted by centrifugation, and resuspended in 80 µL of ice-cold flow cytometry staining buffer (PBS containing 1% FBS). The cells were stained with 5 µL of each fluorescently conjugated antibody or their respective controls for 40 min at 4 °C with occasional shaking. All antibodies were mouse monoclonals, specifically anti-CD 11b (clone M1/70), anti-CD 11c (clone N418), anti-GR-1(clone RB6-8C5), anti-CD3 (clone 12A2), anti-CD4 (clone GK 1.5), anti-CD8 (clone 5367), anti-IL-17 (clone TC11-18H10-1), anti-Ly6c (HK 1.4), anti-CD206(C068C2), and anti-NK1.1 (clone PK136). For intracellular staining, the cells were washed twice with FACS staining buffer and thoroughly re-suspended in BD Cytofix/Cytoperm solution (BD-PharMingen, San Diego, CA, USA) for 20 min. The cells were again washed twice with BD perm/wash solution after storage for 10 min at 4 °C. Intracellular staining for IL-17 and analysis was performed as recommended by the Biolegend protocol. Afterward, cells were washed and resuspended in 300 µL of flow cytometry buffer for analysis. An Agilent NovoCyte Flow Cytometer (Santa Clara, CA, USA) was used to measure the fluorescent signals, which were reported in comparison to the isotype control in each case. 

### 2.6. Western Blot Analysis

Cells from mouse LPLs were lysed, pooled for the appropriate three groups, and the concentration of total protein in each lysate pool was estimated using a BCA assay. A total of 30 µg of lysate from the control, DSS, and SMM-189 groups were separated in 10% SDS-PAGE gels, and the proteins were transferred to a PVDF membrane (#1620174, Bio-Rad, Hercules, CA, USA) using a Trans blot turbo (Bio-Rad, Hercules, CA, USA). The membrane was blocked with intercept blocking buffer (#92760001, LI-COR Biosciences, Lincoln, NE, USA) at room temperature (RT) for 1 h and incubated with mouse primary antibodies anti-CB2 (#SC-293188, Santa Cruz Biotechnology, Dallas, TX, USA; 1:200), anti-PKA (#SC-136231, Santa Cruz Biotechnology; 1:200), anti-β-actin (#926-42212, LI-COR Biosciences; 1:1000) at 4 °C overnight while shaking. After washing, the membranes were incubated with IRDye^®^800CW-conjugated goat anti-mouse secondary antibodies (#926-32210, LI-COR Biosciences; 1:5000) at RT for 1 h. Signal intensity was measured using the LI-COR Biosciences Odyssey Sa Infrared Imaging system, and results were analyzed using Image Studio (Lite Ver 5.2) software.

### 2.7. Histology

Colons were excised from individual mice from 3 independent replicates and rinsed with fresh PBS, fixed in 10% paraformaldehyde, and embedded in paraffin. The fixed tissues were sectioned at 4 μm and stained with hematoxylin and eosin (H&E) in the UTHSC histology core facility. Each section was examined under a microscope, scored in a blinded fashion, and graded according to the severity of the disease. A score (0 to 5) was given based on previously established criteria for colitis developed in our laboratory [24]. The summation of scores provided a total colonic disease score per mouse.

### 2.8. Statistical Analysis

Statistical analysis was performed using Prism 6.0 software (GraphPad Software, San Diego, CA, USA). The 2-tailed unpaired Student t-test was used for statistical comparisons (* *p*  <  0.01, ** *p*  <  0.001).

## 3. Results

### 3.1. SMM-189 Modulates Body Weight and Colon Length in Mice with DSS-Induced Colitis

Reduction in body weight and colon length are key criteria that define the progression of colitis. To induce colitis, mice were administered 3.5% DSS in drinking water for 7 days and examined daily for body weight, stool consistency, and any sign of distress. The mice that received DSS lost up to 13% of their initial body weight compared to mice in the control group that received normal drinking water. However, the mice that were administered DSS plus SMM-189 showed improved body weight compared to DSS administration alone (*n* = 18; *p* < 0.001) (Figure 1A). Further, we observed that colon length, which was shortened in the DSS group relative to the vehicle control, was improved after SMM-189 treatment (*n* = 18; *p* < 0.01) (Figure 1B,C). Taken together, these results suggest that SMM-189 improves body weight and colon length in experimental colitis.

### 3.2. Activation of the CB2 Receptor by SMM-189 Inhibits Infiltration of Th17 Cells

Th17 cells are crucial to the regulation of the immune response during the development of IBD. Alteration of the Th17-mediated immune response, specifically infiltration or dysregulation of Th17 cells, results in colitis. In this study, we examined the infiltration of T cells and Th17 cells in the spleen, MLNs, and LPLs in all three groups of mice. The percentages of CD4^+^ in the spleen were decreased in the DSS group relative to the control group (16.46 vs. 5.72) (*p* < 0.01) (Figure 2A). However, this percentage was increased in the SMM-189-treated group relative to the DSS group (13.36 vs. 5.72; *p* < 0.01). We also analyzed the Th17 cell population in all three groups. The percentage of Th17 in the spleens of mice in the control group (7.13%) increased to 16.07% in mice that received DSS alone (Figure 2C) (*p* < 0.001). Interestingly, the percentage of Th17 in the spleens of mice that received SMM-189 had reduced to 2.93% relative to the DSS group (Figure 2C). The percentage of Th17 cells in the LPLs and MLNs was also reduced significantly in the SMM-189-treated group relative to the group that received DSS alone (*p* < 0.01) (Figure 2C). Taken together, these results suggest that treatment of mice with the CB2 inverse agonist SMM-189 reduces the frequency of Th17 inflammatory cells in the spleen, MLNs, and LPLs, which might be responsible for the suppression of colitis.

### 3.3. SMM-189 Increases the Frequency of Myeloid-Derived Suppressor Cells (MDSCs) in the Spleen and LPLs of Mice with DSS-Induced Colitis

The MDSCs represent a heterogeneous population of immature myeloid cells. They have the remarkable ability to suppress both adaptive and innate immune responses. In mice, MDSCs are characterized by co-expression of the DC marker CD11b with the myeloid-cell lineage differentiation antigen Gr-1. In this study, we observed that the percentage of MDSCs increased in the spleens of mice treated with SMM-189 relative to mice treated with DSS alone (*p* < 0.01) or the vehicle control (*p* < 0.01) (Figure 3A,B). Similarly, we observed a slight decrease in the percentage of MDSCs in the MLNs of mice from the SMM-189 group compared to those in the DSS group (*p* < 0.01) (Figure 3B). However, we observed a significant increase in the frequency of MDSCs in the LPLs of mice that received DSS plus SMM-189 when compared to those that received DSS alone (*p* < 0.01) (Figure 3A,B). Taken together, these data suggest that SMM-189 treatment increased the percentage of MDSCs in the spleen and LPLs, which may play a suppressive role to reduce the inflammation characteristic of DSS-induced colitis.

### 3.4. SMM-189 Differentially Alters the Percentages of NK and NKT Cells during DSS-Induced Colitis

Natural killer (NK) cells are innate lymphocytes that provide immune surveillance against inflammation and help to defeat external threats. To understand the role of NK cells in DSS-induced colitis, we analyzed the relative percentages of NK (NK1.1) and NKT (CD3+NK1.1) cells using flow cytometry analysis. Our results demonstrate that the percentage of NK cells in the MLNs and LPLs increased in the SMM-189-treated group relative to the group that received DSS alone (*p* < 0.01) (Figure 4A,C). In contrast, we observed a decrease in the percentage of NK cells in the spleens of mice treated with SMM-189 relative to those administered DSS alone (*p* < 0.01) (Figure 4C). Further, we observed an increase in the percentage of spleen and LPLs NKT cells in the mice that received SMM-189 relative to the mice administered DSS alone (*p* < 0.01) (Figure 4A,B). However, NKT cells appeared to increase in the spleens of the SMM-189-treated group when compared to DSS alone (*p* < 0.01) (Figure 4B). Taken together, our data suggest that SMM-189 causes differential accumulation or depletion of NK and NKT cells, respectively, in systemic and mucosal organs to suppress inflammation.

### 3.5. SMM-189 Reduces Neutrophil Infiltration

Neutrophils are a major source of inflammatory cytokines and are among the first cell types to arrive at the site of inflammation and are measured to be elevated in IBD patients. Because neutrophils play a key role in experimental models of colitis [25,26], we determined the effect of SMM-189 treatment on neutrophil infiltration compared to mice that received DSS alone. We observed that the percentages of both systemic and mucosal neutrophils were decreased in mice treated with SMM-189 relative to mice treated with DSS alone (*p* < 0.01). (Figure 5A,B). Taken, together these data suggest that during acute experimental colitis in mice, SMM-189 reduces the percentage of neutrophils to attenuate intestinal inflammation.

### 3.6. SMM-189 Treatment Induces Expression of CB2 and Protein Kinase A (PKA)

To determine the molecular mechanism by which SMM-189 suppresses DSS-induced colitis, we investigated whether SMM-189 affects the level of endogenous CB2 receptors in the intestine. Normally, CB2 receptor agonists inhibit PKA activation, but increased cAMP levels associated with CB2 inverse agonists can activate PKA, thus differentially affecting downstream signal transduction pathways. We used Western blot analysis with antibodies specific for CB2 and PKA to determine the relative levels of these proteins in the LPLs of mice from all three groups. Treatment of mice with SMM-189 led to increased expression of CB2 in the LPLs relative to mice that received DSS alone and the control group (Figure 6A,B). We also detected an increase in PKA levels in the LPLs of mice administered SMM-189 relative to those detected in mice administered DSS alone and the control group (Figure 6A,C). Together these findings suggest that SMM-189 may reduce the pro-inflammatory response by increasing the expression of both endogenous CB2 expression and PKA, thereby inducing signal transduction via the PKA pathway and ultimately suppressing DSS-induced colitis.

### 3.7. SMM-189 Treatment Reduces the Severity of Colitis

We also performed histopathological analysis on colon tissue from mice in all three groups to allow us to determine to mean inflammation scores in each case and further evaluate the potentially protective role of SMM-189 during DSS-induced colitis. We detected no histopathological changes in the colon of mice in the control group, and the mean inflammation score provided a baseline level for the absence of DSS-induced colitis (Figure 6D,E). In contrast, we observed increased cellular infiltration, epithelium damage, and crypt distortion in mice administered DSS alone and a correspondingly high mean inflammation score, as expected for DSS-induced colitis (Figure 6D,E). However, in the mice administered DSS and SMM-189, cellular infiltration was reduced and the histological inflammatory scores (*p* < 0.01) for the colon were decreased relative to mice that received DSS alone (Figure 6D,E). In summary, all these results support the hypothesis that SMM-189 increases the expression of endogenous intestinal CB2 and PKA in the intestine, which presumably induces signaling transduction via PKA pathways, and reduces the pathology associated with inflammation, thus also lowering the inflammation score. Therefore, the administration of SMM-189 ultimately reduced the hallmarks of DSS-induced colitis in mice.

## 4. Discussion

Intestinal inflammation initiates immune cell-mediated pathways and signaling cascades that ultimately result in the pathological state of colitis. Dysregulation of immune responses in IBD results in a chronic relapsing-remitting gastrointestinal (GI) tract disease. IBD represents a global economic and social burden, particularly because conventional treatments are not equally effective for all patients and carry the risk of severe side effects. Thus, there is an ongoing need for the development of novel, safe therapeutic options for the treatment of IBD. Previous reports have implicated both CB1 and CB2 receptors as potential targets for the development of IBD therapies [22,27,28,29,30] and ligands that bind to these receptors can relieve patients’ IBD symptoms [31]. To the best of our knowledge, the effect of a CB2 inverse agonist on experimental colitis has not been reported to date. Therefore, we tested the effect of the CB2 inverse agonist SMM-189 on the DSS-induced experimental colitis model in female WT C57BL/6 mice and demonstrated that SMM-189 suppressed DSS-induced colitis in these animals. SMM-189 treatment alleviated the loss of body weight, lowered the inflammatory disease score, and reduced the severity of the disease. We found that SMM-189 treatment reduced the percentage of mucosal and systemic Th17 cells and neutrophils, which increased during colitis and increased the percentage of MDSCs and NKT cells. SMM-189 treatment also led to increased protein kinase A (PKA), which presumably increased signal transduction in PKA pathways. Taken together, our results suggest that SMM-189 abrogates colitis by suppressing the activation of Th17 cells and neutrophils and increasing the frequency of MDSCs and NKT cells. Furthermore, SMM-189 also increased expression levels of CB2 receptors and PKA, thereby potentially reducing the numbers of pro-inflammatory immune cells that accumulate at sites of inflammation in the colon to reduce the symptoms of colitis.

Cannabinoids have been used as therapeutic agents for experimental models of colitis and to treat inflammation and alterations in mood and memory. For the 2,4,6-trinitrobenzene sulphonic acid (TNBS)-induced model of acute colitis in rats, treatment with delta-9-tetrahydrocannabinol (THC) and the CB2 agonist AM 630 suppressed inflammation and reduced expression of CB2 receptor mRNA [22,32]. Mice that genetically lacked CB1, CB2, or both, exhibited aggravated inflammation in the TNBS-induced colitis model [33], but blocking the degradation of endocannabinoids inhibited inflammation and reversed the shortening of the colon [29]. Clinically, endocannabinoids are found at increased levels in biopsies from human UC patients [34]. Our previous work revealed that the CB2 agonist JWH-133 ameliorates spontaneous colitis by the attenuation of activated T cells [21]. The current study demonstrates for the first time that the CB2 inverse agonist SMM-189 can restore body weight and suppress colitis symptoms. It has been shown that the neutral CB receptor antagonists AM4113 and AM630 reduce body weight by reducing food intake [35,36]. The data at our disposal does not support the previous findings, because we did not measure the food intake in these mice. However, we noticed that SMM-189 mediated improvement in the body weight of mice suffering from DSS-induced colitis.

CB2 receptors are expressed on immune cells and targeting CB2 receptors on T cells, macrophages, neutrophils, and NKT cells may provide an opportunity for the development of novel therapeutics for colitis. T cells play a crucial role in IBD progression since levels of activated CD4^+^ T cells are higher in both IBD and experimental models of chronic colitis. The Th17 subset of T cells produces IL-17 after differentiation and plays a key role in the pathogenesis of chronic intestinal inflammation [37]. Th17 cells also play important roles in the induction of experimental colitis in mice [38]. In this study, we determined that treatment of DSS-induced mice with SMM-189 altered the percentage of both systemic and mucosal Th17 cells in the spleen, MLNs, and LPLs relative to mice that received DSS alone. Activation of CB2 receptors selectively reduces the production of IL-17 by Th17 lymphocytes during fibrogenic responses in the liver [39] and experimental autoimmune encephalomyelitis (EAE) [40,41]. Thus, our findings are corroborated by previous studies using various autoimmune disease models. Taken, together our data suggest that impairment of the Th17 response in mice by SMM-189 treatment is associated with protection from DSS-induced colitis.

MDSCs represent a key myeloid precursor population of innate immune cells that suppress inflammatory responses [42]. Both humans and mice with colitis exhibit a dramatic increase in the number of MDSCs, particularly those of the granulocytic subset (G-MDSC) [8]. Interestingly, activation of cannabinoid receptors leads to a massive accumulation of MDCSs with immunosuppressive properties [43]. We showed here that mice administered DSS plus SMM-189 exhibit increased percentages of systemic and mucosal MDSCs in the spleen, MLNs, and LPLs relative to mice that received DSS alone and the control group. We hypothesize that the increased percentage of MDSCs after SMM-189 treatment is most probably responsible for the suppressed immune response in the colon, leading to a decrease in inflammation and the suppression of colitis. However, more detailed studies are needed to strengthen the mounting evidence that MDSCs induced or recruited by SMM-189 treatment are responsible for the suppression of colitis. One such experiment might explore the effect of the adoptive transfer of proliferating MDSCs to an acute or spontaneous colitis model in mice.

IBD is associated with an influx of neutrophils to the mucosa, which plays important role in mucosal injury and inflammation. Antibody-mediated depletion of neutrophils suppresses several features in the various experimental models of colitis [26,44], and neutrophil motility protects the GI tract from IBD after curcumin treatment [45]. NKT cells express markers for both conventional T cells and NK cells, participate in the prevention of autoimmune disease, and play both protective and destructive roles in various experimental models of colitis [46,47]. In this study, mice treated with DSS plus SMM-189 exhibited significantly diminished percentages of neutrophils relative to those detected in mice treated with DSS alone. This is in contrast with a report that activation of the CB2 receptor by JWH-133, a CB2 agonist, protects against cerebral ischemia mainly by inhibiting the recruitment of neutrophils [48]. This may reflect differences in the response to CB2 agonists and inverse agonists in the central nervous system versus the systemic immune cell response [49,50]. NK cells also protect mice from DSS-induced colitis by regulating neutrophil functions, in this case via NKG2A receptors [51]. NKT cells also inhibit cells of the Th17 lineage, which is associated with colitis and several autoimmune diseases. Thus, the increase in the percentage of NKT cells in the LPLs after treatment of mice with DSS plus SMM-189 probably suppresses colitis by regulating the functions of both Th17 cells and neutrophils. Our study supports previous findings reported in the literature and suggests that SMM-189 may suppress levels of neutrophils and Th17 cells in part by the induction of NKT cells, thereby attenuating chronic colitis.

We next explored the molecular mechanism by which SMM-189 suppresses colitis. In this study, we observed that the administration of SMM-189 to DSS-induced mice enhanced the endogenous expression of CB2 receptors and PKA in the LPLs relative to the level observed in the DSS and control groups. It has been shown that enhanced expression of endogenous CB2 receptors and PKA play a critical role in epithelial barrier recovery and PKA-dependent inhibition of several signaling pathways, leading to anti-inflammatory effects [52,53]. Furthermore, previous work from other groups also showed that a CB2 inverse agonist increased cAMP levels that further enhanced PKA expression [54]. Elevated levels of PKA can inhibit NF-kB pathways, hence exerting anti-inflammatory activity. Hence, we hypothesize that SMM-189 may trigger the expression or enhance the endogenous expression of CB2 receptors, which further activates the PKA pathway to reduce inflammation in the DSS-induced model of colitis.

In summary, the results of the current study suggest that SMM-189 might serve as an anti-inflammatory agent by targeting Th17/MDSC/NKT/neutrophil pathways during the development of DSS-induced colitis. Further, SMM-189 induces the expression of endogenous CB2 receptors and PKA and may induce signaling via PKA pathways. While limited studies on CB2-receptor agonist-mediated inflammation have been reported, this study shows that the CB2 inverse agonist SMM-189 also suppresses experimental colitis, most likely through inhibition of Th17 cells and neutrophils, and induction of MDSCs, NKT cells, and anti-inflammatory PKA signaling pathways. Our previous studies have shown that SMM-189 can be used in clinical studies to treat central nervous system diseases [49] and might serve as a unique probe in diseases like colitis to evaluate the signaling mechanisms associated with CB2 inverse agonist anti-inflammatory activity. However, more detailed mechanistic studies on how CB2 agonists induce PKA signaling pathways are required for a prudent conclusion.

## Figures and Tables

**Figure 1 pharmaceutics-14-00936-f001:**
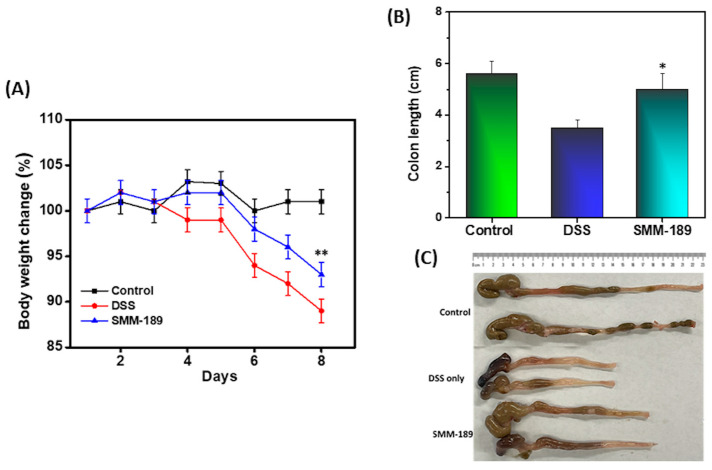
Body weight and colon length alteration after DSS induction of colitis in the absence or presence of SMM-189 treatment. Normal C57BL6 mice received distilled water (■), DSS alone in drinking water (
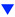
), or DSS plus 100 µL of SMM-189 (
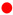
) every day, starting at the beginning of day 0. The body weight of the mice was recorded every day, and the change from initial body weight was expressed as a percentage change in body weight Panel (**A**). On day 9, the colon length was recorded after euthanization Panel (**B**). Panel (**C**) Represent the picture of colon length of mice taken from individual groups. The statistical significance between values of each group was assessed using Student’s *t*-test. Data represent the mean of three independent experiments involving six mice per group (*n* = 18). Asterisks (*) (**) indicate a statistically significant difference at *p* < 0.01 and *p* < 0.001, respectively, between the group of mice treated with vehicle (DSS) and SMM-189.

**Figure 2 pharmaceutics-14-00936-f002:**
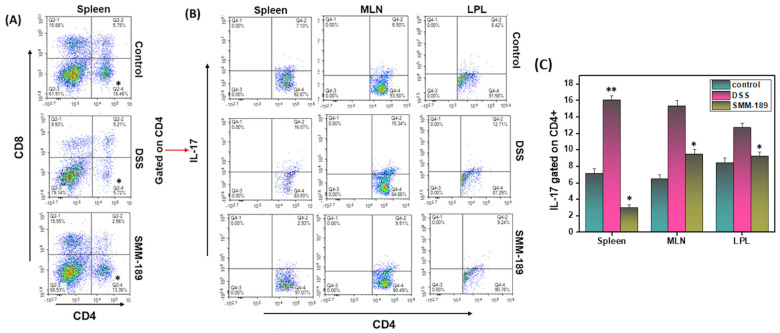
**SMM-189 alters T cells and Th17 phenotypes during DSS-induced colitis**. Spleens, MLNs, and LPLs were harvested from each of the three groups of C57BL6 mice described in the legend for Figure 1 and single-cell suspensions were prepared for each tissue. The cells were stained with antibodies specific for CD4, CD8, and IL-17 and analyzed by flow cytometry. Panel (**A**), the frequency in the lower right quadrant indicates the total percentage of CD4^+^ T cells and the upper left quadrant indicates the total percentage of CD8^+^ T cells. Panel (**B**), the frequency in the upper right quadrant indicates the total percentage of Th17 cells gated on CD4^+^ T cells. Panel (**C**), the percentage of total Th17 cells (*n* = 18; mean of six samples ± SEM) in the spleens, MLNs, and LPLs of each group was determined. Data are shown for a representative one of three independent experiments. Asterisks (*) (**) indicate a statistically significant difference at *p* < 0.01 and *p* < 0.001, respectively, between the group of mice treated with vehicle (DSS) and SMM-189.

**Figure 3 pharmaceutics-14-00936-f003:**
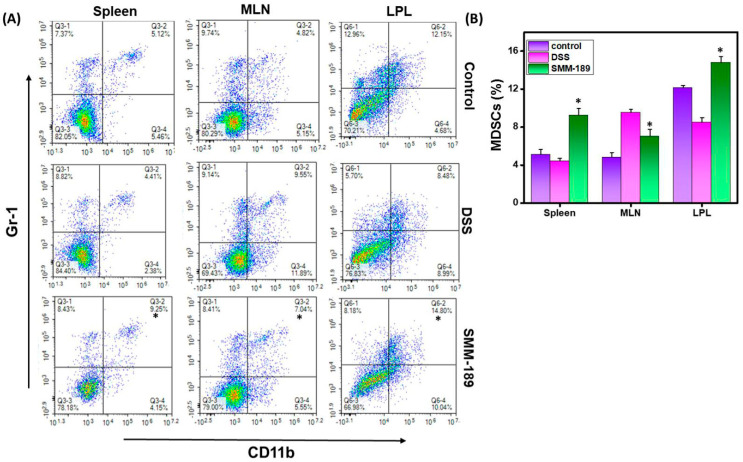
**SMM-189 induces MDSCs during DSS-induced colitis.** Spleens, MLNs, and LPLs were harvested from the three groups of mice described in the legend in Figure 1 and single-cell suspensions were prepared from each tissue. Cells were stained with antibodies specific for the DC marker CD11b and myeloid-cell lineage differentiation marker Gr-1 and analyzed by flow cytometry. Panel (**A**), changes in the frequency of MDSCs (CD11b^+^ GR-1^+^) in the spleen, MLNs, and LPLs were compared for the control, DSS, and SMM-189-treated groups. In panel (**B**), the numbers shown in the upper left quadrant of the plots in panel (A), indicating the total percentage of MDSCs, are shown in the bar graph. Data represent the total number of cells ± SEM from three independent experiments each involving six mice per group (*n* = 18). Asterisks (*) indicate statistically significant differences at *p* < 0.01 between the DSS and SMM-189-treated groups.

**Figure 4 pharmaceutics-14-00936-f004:**
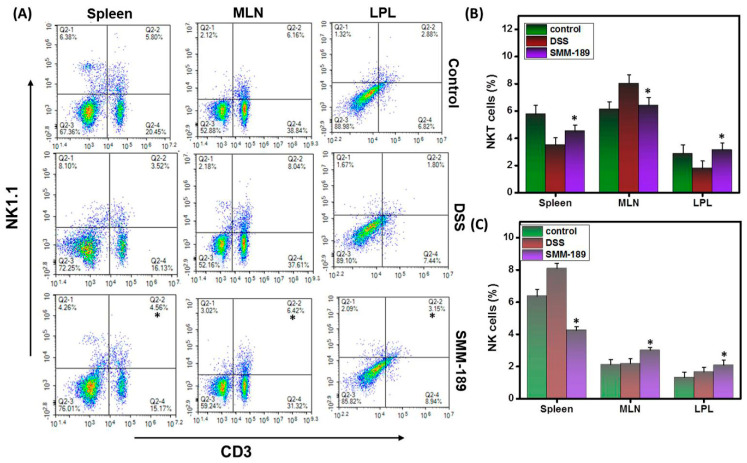
**Alteration in NKT and NK cells after SMM-189 treatment during DSS-induced colitis.** Spleens, MLNs, and LPLs were harvested from the three groups of mice described in the legend for Figure 1 and single-cell suspensions were prepared from each tissue. Cells were stained with antibodies specific for T cell marker CD3 and natural killer cell marker NK1.1. Panel (**A**), changes in the mean percentage of NK and NKT cells relative to CD3^−^ or CD3^+^ T cells from spleen, MLNs, and LPLs were compared between the DSS and SMM-189-treated groups. In Panel (**B**), the numbers from the upper right quadrants of the plots in panel (**A**), indicating the total percentage of NKT cells, were collected for three experiments in a bar graph. Panel (**C**), the numbers from the upper left quadrants of the plots in Panel (**A**), indicating NK cells, were collected for three experiments in a bar graph. Data represent the total number of cells ± SEM from three independent experiments, each involving six mice per group (*n* = 18). Asterisks (*) indicate statistically significant differences at *p* < 0.01 between the DSS and SMM-189-treated groups.

**Figure 5 pharmaceutics-14-00936-f005:**
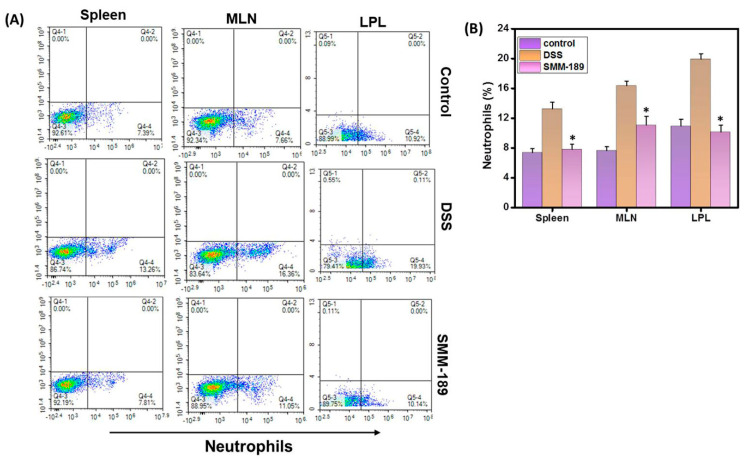
**SMM-189 treatment reduced the percentage of neutrophils during DSS-induced colitis.** Spleens, MLNs, and LPLs were harvested from the three groups of mice described in the legend for Figure 1 and single-cell suspensions were prepared from each tissue. Cells were stained with antibodies specific for the T cell marker CD3 and the neutrophil marker LY6G. Panel (**A**), CD3^−^ lymphocytes from the spleen, MLNs, or LPLs were screened for the presence of neutrophils (LY6G) by flow cytometry. The percentage in the lower right quadrants of the plots in panel (**A**) reflects the total neutrophils and is shown in the bar graph of Panel (**B**). Data represents the total percentage of cells ± SEM from three independent experiments, each involving six mice per group (*n* = 18). Asterisks (*), indicate statistically significant differences (*p* < 0.01) between the vehicle (DSS) and SMM-189-treated groups.

**Figure 6 pharmaceutics-14-00936-f006:**
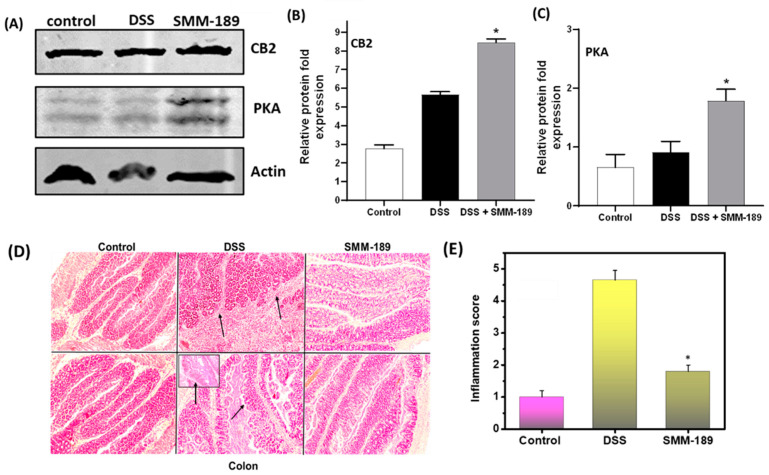
**Induction of CB2, PKA, and histological alteration after SMM-189 treatment.** LPLs and colon tissue were harvested from the three groups of mice in this study. (**A**) represents the immunoblot analysis of CB2 and PKA. (**B**) represents the relative fold expression of CB2 and (**C**) represents the relative fold expression of PKA. (**D**) representative histological sections of colons from the three groups of mice. DSS-treated mice that did not receive SMM-189 treatment exhibited significant lymphocyte infiltration and distortion of crypts (**central** panel arrow), but SMM-189-treated mice exhibited markedly decreased lymphocyte infiltration (**right** panel); these changes were not observed in tissue from mice in the control group (**left** panel). Other changes include diffuse leukocyte infiltrates and thickening of the lamina propria near the distorted crypts in the colon. (**E**) represents the combined inflammation scores, which were reduced for the SMM-189-treated group compared to DSS alone. Representative sections from three separate experiments (10X magnification) are shown, with each group containing six mice (*n* = 18). Asterisks (*) indicate statistically significant differences at *p* < 0.01 between the vehicle (DSS) and SMM-189-treated groups.

## Data Availability

The raw data supporting the conclusions of this article will be made available by the authors, without any reservation.

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
