# Peer review of "Cannabinoid Receptor 2 (CB2) Inverse Agonist SMM-189 Induces Expression of Endogenous CB2 and Protein Kinase A That Differentially Modulates the Immune Response and Suppresses Experimental Colitis"

_pharmaceutics, 2022, doi:10.3390/pharmaceutics14050936_

Round 1

Reviewer 1 Report

The manuscript entitled “Cannabinoid receptor 2 (CB2) inverse agonist SMM-189 induces expression of endogenous CB2 and protein kinase A that differentially modulates the immune response and suppress experimental colitis” is a good attempt on the part of authors to summarize effect of SMM-189 on experimental colitis. The article though well written needs a few rectifications to add on the overall impact. I have a sincere query to know why a research article like this been selected as review article? If it’s a mistake then please change the category of submission of the article. A few suggestions to the authors that can be made to improve overall scientific quality of the well written manuscript.

  1. Keywords can include ‘inflammatory bowel disease’
  2. IFN-A is to be written as IFN-alpha or IFN-α
  3. It seems that specific symbols such α, µ etc. are omitted due to some technical error. Authors are suggested to take care of that.
  4. The numbering of headings in ‘Results’ section is erroneous
  5. In Figure 1 B, has the statistics for colon length of SMM-189 done with respect to control or DSS group?
  6. There is some error in writing the relative values of CD4+ and CD8+ in control vs DSS and control vs SMM-189. Please ensure to double check in the text (Figure 2A).
  7. In Figure 2C, as stated, Th 17 cells number is reduced in SMM-189 treated mice compared to control. What probable explanation is there to account for such intense reduction in treatment group?
  8. The authors claim that “we observed a slight increase in the percentage of MDSCs in the MLNs of mice in the SMM-189 group as compared to those in the DSS group.” But in Figure 3B, the graph is showing decrease of MDSCs in the MNLs in SMM-189 group compared to DSS group. Please ensure that the values incorporated are correct and the authors are suggested to make changes accordingly.
  9. Under the heading “SMM-189 differentially alters the number of neutrophils, NK, and NKT cells during DSS induced colitis”, figure numbers incorporated in the text do not match the one in figure legends.
  10. In the figure 6 legend, labelling is not done correctly. Please ensure to recheck and do the required corrections in the text and/or figure legends wherever necessary to avoid any difficulty in understanding by the reader.
  11. In the ‘Discussion’ section, authors at one point claimed that “support to the hypothesis that SMM-189-mediated improvement in the bodyweight of DSS-induced colitis may also result from reduced food intake.” How reduced intake of food is helpful in improved body weight? Food intake is reduced in DSS treated group and it is visible as body weight is significantly reduced. But for SMM-189 group, it is not very clear.

Author Response

File attached

Reviewer 2 Report

The manuscript by Kiran et al studies the effect of Cannabinoid receptor 2 agonist SMM-189 on the protection against experimental colitis. The results of this manuscripts are very clear and supported by the experimental data. I believe it should be published in the present form.

Author Response

File attached.

Reviewer 3 Report

High quality presenation. Very interesting.

Author Response

File attached.

Reviewer 4 Report

The manuscript is interesting, well organized.
the authors could add in the introduction some manuscripts that also concern CB1 to better clarify the role of cannabinoid receptors in the intestine. (see below).
Martínez V, Iriondo De-Hond A, Borrelli F, Capasso R, Del Castillo MD, Abalo
R. Cannabidiol and Other Non-Psychoactive Cannabinoids for Prevention and Treatment of Gastrointestinal Disorders: Useful Nutraceuticals? Int J Mol Sci. 2020 Apr 26; 21 (9): 3067.

Pagano E, Capasso R, Piscitelli F, Romano B, Parisi OA, Finizio S, Lauritano A, Marzo VD, Izzo AA, Borrelli F. An Orally Active <i> Cannabis </i> Extract with High Content in Cannabidiol attenuates Chemically- induced Intestinal Inflammation and Hypermotility in the Mouse. Front Pharmacol. 2016 Oct 4; 7: 341
Do the authors have information on toxicity?
Do the authors think the microbiota can affect the effectiveness of SMM-189?

In the Discussion, the Authors should highlight the possible clinical significance of their findings

Author Response

File attached.